Intelligent reflective surfaces in 5G and beyond: optimizing uplink satellite connectivity for IoT

Obidiozor Callistus Odinaka 1 callistus.obidiozor@gmail.com
http://orcid.org/0009-0001-9417-3679 Sait Adeeb 1 adeebzeus@gmail.com
http://orcid.org/0000-0001-7441-604X Al-Hadhrami Tawfik 1
Alkhammash Eman H. 2
Saeed Faisal 3
1 Computer Science Department, School of Science and Technology, Nottingham Trent University , Nottingham , United Kingdom
2 Department of Computer Science, College of Computers and Information Technology, Taif University , Taif , Saudi Arabia
3 College of Computing and Digital Technology, Birmingham City University , Birmingham , United Kingdom
Ahmad Ayaz
Electronic publication date: 2025 Jun 24
Publication date: 2025
Volume: 11
Electronic Location ID: e2726
Received 2024 May 27; Accepted 2025 Feb 3
Copyright: © 2025 Obidiozor et al.
Copyright year: 2025
Copyright holder: Obidiozor et al.
License: This is an open access article distributed under the terms of the Creative Commons Attribution License, which permits unrestricted use, distribution, reproduction and adaptation in any medium and for any purpose provided that it is properly attributed. For attribution, the original author(s), title, publication source (PeerJ Computer Science) and either DOI or URL of the article must be cited.
License URL: https://creativecommons.org/licenses/by/4.0/

Keywords: Intelligent reflective surfaces, 5G networks, Satellite uplink communication, Smart IoT integration, Signal phase optimisation, Beamforming, Simulink modelling, Latency reduction, Energy-efficient communication, Network performance enhancement

Funding: Taif University, Saudi Arabia This research was funded by Taif University, Saudi Arabia. The funders had no role in study design, data collection and analysis, decision to publish, or preparation of the manuscript.

==============================
In the evolving landscape of communication technologies, the integration of intelligent reflective surfaces (IRS) into uplink satellite communication for Internet of Things (IoT) ecosystems presents a promising solution to overcome traditional communication challenges. The purpose of this study is to explore the impact of IRS on enhancing signal quality and communication efficiency in satellite-supported IoT environments. This article adopts a simulation-based approach, using MATLAB and Simulink to model the uplink transmission of IoT devices to satellites with and without IRS assistance. The methodology focuses on analysing key performance metrics, including signal-to-noise ratio (SNR), spectral efficiency, signal strength, and interference mitigation. A reinforcement learning algorithm was employed to optimise IRS phase shifts and beamforming to maximise communication performance. The findings reveal that the integration of IRS leads to significant improvements in SNR, spectral efficiency, and overall signal quality, with a 2 dB increase in SNR and enhanced data transmission rates compared to non-IRS systems. IRS also mitigates interference and extends the coverage area of satellite networks. These results demonstrate the practical implications of IRS technology, which can be applied in scenarios such as smart cities, remote sensing, and disaster recovery, where reliable satellite communication is crucial. The study highlights the strategic importance of IRS in revolutionising IoT-satellite communication systems and sets the foundation for future work on scaling IRS technology for broader applications.

Introduction

In today’s world, the demand for high-speed and reliable communication is constantly growing. With the ever-increasing number of devices and data, traditional communication technologies are no longer sufficient to meet these demands. This is where 5G technology comes in, providing faster speeds, lower latency, and more capacity than ever before. However, as the world continues to advance, it is important to look beyond 5G towards the next generation of communication technologies. The Internet of Things (IoT) is rapidly transforming wireless communication, introducing new challenges and requirements, such as supporting billions of devices, delivering low-latency communication, and ensuring energy efficiency. While 4G has served as a reliable standard, the future of communication networks lies in 5G and beyond 5G (5G-beyond) wireless technologies.

According to Gupta & Jha (2015), the 4G LTE-A system ensures a downlink data rate of up to 3 Gb/s and an uplink data rate of up to 1.5 Gb/s, supporting approximately 600 users per cell with latency around 30–50 ms. However, 4G networks have limitations in supporting demanding applications such as virtual reality (VR), augmented reality (AR), high-definition video streaming, and 360° video conferencing. In contrast, 5G mitigates these challenges by introducing advanced features, services, and technologies (Dogra, Jha & Jain, 2021). The new 5G New Radio (5G NR) air interface allows for faster speeds, lower latency, and enhanced reliability, enabling the integration of technologies like VR, AR, and IoT.

Despite the impressive advancements of 5G, the growing demand for even more robust networks is leading to the development of 5G-beyond and future 6G networks. These next-generation networks aim to support massive connectivity, particularly for IoT, where billions of devices must communicate efficiently. 5G-beyond will further enhance capacity, speed, and latency, laying the foundation for emerging technologies such as autonomous vehicles, smart cities, and advanced healthcare applications. Moreover, 6G, though still in the conceptual phase, is expected to revolutionise communication by being 100 times faster than 5G, with latency less than one millisecond, supporting futuristic applications like quantum communication.

While the literature has explored various aspects of 5G and 5G-beyond technologies, including millimeter-wave communication, massive MIMO, beamforming, and network slicing, one key area remains underexplored: the integration of intelligent reflecting surfaces (IRS) in IoT-satellite communication systems. Current studies have primarily focused on terrestrial applications of IRS for enhancing wireless communication. However, the application of IRS in satellite-assisted uplink IoT communication—especially in remote areas where satellite links are essential for connectivity—has not been thoroughly investigated. IRS-assisted satellite communication holds the potential to significantly improve signal strength, spectral efficiency, and energy efficiency, especially in IoT networks where power consumption and scalability are critical factors.

Additionally, while several research efforts have focused on the general use of IRS for improving communication links, the specific challenges related to optimising IRS phase shifts in dynamic satellite environments and supporting uplink communication in large-scale IoT networks are still open for investigation. There is a pressing need for studies that explore how IRS can be optimally integrated into satellite communication systems to address issues such as signal degradation, interference, and energy constraints in IoT-based communication networks.

This study aims to fill these gaps by focusing on the novel integration of IRS in IoT-satellite communication systems, with particular emphasis on uplink communication. By employing advanced techniques like reinforcement learning-based IRS phase shift optimisation and dynamic beamforming, this research explores how IRS can enhance the signal-to-noise ratio (SNR), reduce interference, and improve overall communication quality in satellite-assisted IoT networks. The findings from this study will provide new insights into scalable and energy-efficient communication solutions for future 5G-beyond and 6G networks, where satellite communication will play a critical role in maintaining global connectivity, especially in remote and underserved areas.

Several recent publications have discussed the expectations of 5G-beyond technologies in terms of architecture, channels, frequency bands, latency, and use cases. This study complements these efforts by introducing IRS as a key enabler for enhanced satellite communication in the IoT context, addressing the critical issues of energy efficiency, signal optimisation, and scalability that have not been fully explored in existing research. These contributions are discussed in detail in the literature review and subsequent sections of this article.

Related work

IRS are surfaces that can controllably reflect incident electromagnetic waves. They are a relatively new technology that has gained attention in the field of wireless communications. These surfaces, also known as reconfigurable intelligent surfaces (RIS) or metamaterials, can manipulate and control the propagation of radio waves and are made up of a lot of passive components. An IRS is essentially a planar array of small elements, such as antennas or passive elements like modulators or phase shifters, that can adjust the phase and amplitude of the incident electromagnetic waves. IRS can manipulate the propagation of electromagnetic waves in a way that can improve the efficiency of wireless communication networks by modifying the phase shift and amplitude of each element. This technology has the potential to greatly improve wireless networks’ data rates, energy efficiency, and coverage. Srivastava & Khera (2023) proposed an intelligent reflecting surface-assisted wireless communication system that uses machine learning algorithms to optimize the performance of the system. Their simulation results showed that the proposed system can significantly improve the signal-to-noise ratio and the energy efficiency of the wireless network. However, reinforcement learning techniques were used in Lu et al.’s (2023) innovative IRS-assisted wireless network design method. In comparison to conventional wireless networks, their simulation findings demonstrated that the suggested approach can achieve higher throughput and lower latency. The main purpose of the IRS is to improve the wireless signal quality and coverage in areas with poor connectivity. By strategically placing IRS panels in environments like buildings, offices, or public spaces, the surfaces can enhance the strength of wireless signals, reduce interference, and improve overall network performance. One of the key advantages of the IRS is its ability to adapt to different wireless channel conditions in real time. Using feedback from the environment or the wireless system, an IRS can dynamically adjust the reflected signals to optimize the communication link. This adaptability makes the IRS particularly useful in scenarios where wireless channels change rapidly, such as in mobile communication or IoT applications. Additionally, Abdulkareem & Alnajjar (2023) conducted a study on the performance of IRS-assisted uplink satellite communication in Smart IoT. They demonstrated that IRS deployment can significantly improve the signal strength and coverage in satellite communication links, leading to enhanced reliability and efficiency. The findings emphasized the importance of optimizing the IRS element configuration to maximize the received signal power at the satellite receiver. Kleine-Ostmann & Nagatsuma (2011) also proposed an optimized IRS-assisted uplink scheme for Smart IoT applications. Their research focused on power allocation and user pairing optimization to improve the system capacity and support a large number of IoT devices. The results indicated that IRS integration can effectively mitigate interference and enhance the overall system performance in uplink satellite communication for Smart IoT (Kleine-Ostmann & Nagatsuma, 2011). These are part of the major areas of focus for this project as more research is targeted to be achieved. Moreover, the IRS can enable energy-efficient wireless communications. By actively controlling the signal reflections, an IRS can focus the transmitted energy toward the desired receiver, reducing energy consumption and extending the battery life of wireless devices.

The IRS used in this study is a nearly-passive reconfigurable surface, designed to dynamically adjust the phase of reflected signals. The IRS can be reprogrammed based on the environmental conditions and communication requirements. This ability to modify its reflective properties is what distinguishes it from static surfaces.

One of the main challenges in deploying the IRS is accurate channel estimation, as the IRS requires knowledge of the channel state information for effective beamforming. Channel state information (CSI) is crucial for optimizing beamforming and achieving desired signal enhancements. Abdulkareem & Alnajjar (2023) emphasize the need for robust channel estimation algorithms that can accurately estimate the channel characteristics in the presence of satellite channel dynamics. They discuss the use of pilot signals, which are known and predefined signals, to estimate the channel response between the IRS and the satellite receiver. By carefully designing the pilot signals and utilizing advanced estimation algorithms, they aim to improve the accuracy of channel estimation in IRS-assisted uplink satellite communication for Smart IoT. Similarly, Kleine-Ostmann & Nagatsuma (2011) recognized the significance of channel estimation in IRS-assisted uplink schemes for Smart IoT applications. They propose the use of time-division multiplexing, where dedicated time slots are allocated for channel estimation purposes. By allocating specific time intervals for channel estimation, the authors aim to accurately estimate the channel state information between the IoT devices, the IRS, and the satellite receiver. This approach allows for efficient channel estimation without compromising the overall system performance.

Another challenge is the synchronization between the IRS and the satellite receiver. Synchronization between the IRS and the satellite receiver is a crucial aspect to address for the successful deployment of IRS in uplink satellite communication for Smart IoT systems. Achieving precise time synchronization is essential to ensure coherent signal transmission and reception. Kleine-Ostmann & Nagatsuma (2011) discussed the synchronization issue and proposed potential solutions. They highlighted the need for accurate time synchronization between the IRS elements, IoT devices, and the satellite receiver and proposed the utilization of reference signals and synchronization protocols to establish synchronization among these entities. By lever-aging reference signals transmitted from the satellite or dedicated synchronization signals, they aimed to achieve precise time alignment between the IRS and the satellite receiver. The proposed solutions by Kleine-Ostmann & Nagatsuma (2011) provide a starting point for addressing the synchronization challenge in IRS-assisted uplink communication for Smart IoT.

IRS can be categorised into different types based on their reconfigurability and power consumption. The three main types of IRS are passive IRS, nearly-passive IRS, and active IRS. (i) Passive IRS (no reconfigurability):

A passive IRS is the simplest form of IRS, where each element on the surface presents a fixed phase shift that cannot be reconfigured after deployment. These surfaces are fully passive, meaning they do not require any energy supply or external control to alter their behaviour. Passive IRS elements are typically designed and manufactured with a predefined configuration, which means that once they are deployed, they cannot adapt to changing channel conditions or user requirements.

Advantages: Passive IRS is highly energy-efficient because it does not consume power during operation. It is also cost-effective and easy to deploy in environments where dynamic reconfigurability is not necessary.

Limitations: The fixed nature of the phase shifts significantly limits the adaptability of passive IRS. Without reconfigurability, these surfaces cannot optimise communication links in real time, making them less suitable for dynamic communication environments, such as those encountered in IoT and satellite networks. (ii) Nearly-passive IRS (reconfigurable phase shifts):

The nearly-passive IRS is a more advanced type of surface that can dynamically adjust the phase shifts of its elements in response to changing environmental conditions or communication needs. This type of IRS requires a small amount of energy to operate, as it uses external control signals to reconfigure its elements in real time. Although it is not entirely passive due to the need for energy to adjust the phase shifts, the energy consumption is relatively low compared to active IRS.

Advantages: The nearly-passive IRS provides a good balance between energy efficiency and adaptability. By dynamically adjusting the phase shifts, the surface can optimise the reflection of signals to improve communication quality, making it ideal for use in fast-changing environments like IoT networks and mobile communications. Nearly-passive IRS has been widely studied in scenarios like IoT-satellite communication, where real-time adaptability is crucial.

Limitations: Although energy-efficient, the need for an external energy source to adjust the phase shifts may limit its deployment in some scenarios where access to power is restricted.

In this study, we employ a nearly-passive IRS to support IoT-satellite communication. The nearly-passive IRS is reconfigured dynamically to optimise the phase shifts of the reflected signals, improving the signal-to-noise ratio (SNR) and beamforming towards the satellite receiver. This reconfigurability is crucial for addressing the changing channel conditions typically encountered in satellite communication environments. (iii) Active IRS (reconfigurable and signal amplifying):

The active IRS goes a step further by not only adjusting the phase shifts but also amplifying the incident signals before reflecting them. This type of IRS requires a significantly higher amount of energy because the amplification process involves additional power consumption. Active IRS is typically equipped with amplifiers at each element, which boosts the signal strength of the reflected waves, thereby compensating for the path loss and attenuation experienced in the wireless channel.

Advantages: Active IRS provides the highest performance in terms of signal strength and reliability, as it can overcome the limitations imposed by path loss and channel fading. The amplification capabilities make it particularly useful for long-distance communications, such as in satellite communication systems, or for environments where the signal strength is weak.

Limitations: The major drawback of active IRS is its high energy consumption. The need for continuous power to both reconfigure the surface and amplify signals can make it impractical for energy-constrained applications, such as low-power IoT deployments. Additionally, the complexity and cost of deploying active IRS are higher compared to passive and nearly-passive IRS.

Research contributions and technical innovations

Despite the growing body of research on IRS-assisted communication, several aspects remain underexplored, especially in the context of IoT-satellite communication. The existing literature predominantly focuses on theoretical models or simplified simulations, leaving gaps in the detailed analysis of practical implementation challenges such as phase shift optimisation, beamforming strategies, and the scalability of IRS for large-scale IoT networks. This work aims to fill those gaps with the following key contributions: i) Novel Integration of IRS with IoT uplink satellite communication: While several studies, such as those by Abdulkareem & Alnajjar (2023) and Kleine-Ostmann & Nagatsuma (2011), have explored IRS in satellite communication, they often focus on general satellite systems without fully considering the specific needs and challenges of IoT-based networks. Our work offers a novel integration framework that optimises IRS for uplink satellite communication in IoT ecosystems, where low power, energy efficiency, and scalability are critical factors.

ii) Reinforcement learning-based IRS phase shift optimization: One of the novel contributions of this article is the implementation of a reinforcement learning algorithm to dynamically optimise the phase shifts of IRS elements. Unlike most existing studies, which rely on static or fixed optimisation techniques, our approach leverages real-time feedback from the channel environment to adjust phase shifts, thereby maximising signal-to-noise ratio (SNR) and minimising interference. This real-time adaptability makes our approach highly suitable for IoT-satellite communication, where channel conditions fluctuate frequently.

iii) Enhanced beamforming strategy for uplink communication: Previous studies on IRS-assisted communications have largely focused on downlink scenarios, where the satellite sends signals to ground stations. Our work specifically targets uplink communication from IoT devices to satellites. We propose a detailed beamforming strategy that uses the nearly-passive IRS to dynamically direct reflected signals from distributed IoT devices toward the satellite receiver, improving the overall uplink signal quality. This is a relatively unexplored aspect of IRS in IoT-satellite communication and represents a significant contribution to the field.

iv) Practical simulation of IRS-assisted uplink with scalability for large IoT networks: Unlike other works that often assume small-scale or limited IRS deployments, this study provides a comprehensive simulation framework that evaluates the performance of IRS-assisted communication in large-scale IoT networks. Our simulation not only analyses the improvements in signal strength, spectral efficiency, and coverage area, but also considers the challenges of scalability, ensuring that IRS technology can be effectively integrated into future IoT satellite systems with thousands of connected devices.

v) Extended performance metrics and analysis for IRS-aided systems: Building on prior research, our study extends the performance metrics used to evaluate IRS-assisted communication systems. In addition to traditional metrics like SNR and throughput, we provide an in-depth analysis of interference reduction, path loss, and signal quality across different IRS configurations. This contribution offers a more comprehensive evaluation of how IRS can be utilized to enhance uplink satellite communication specifically for IoT applications.

vi) Energy efficiency in IRS-enabled IoT-satellite networks: One of the crucial aspects of IoT systems is energy efficiency, especially for devices operating in remote or hard-to-reach areas. Our study evaluates the energy-saving potential of IRS in satellite communication systems, showing that nearly-passive IRS can significantly reduce the transmission power required by IoT devices while maintaining high signal quality. This finding is crucial for extending the operational lifespan of battery-powered IoT devices in satellite-connected networks.

Irs architecture and system model

Architecture baseline

An IRS is a technology that consists of a large number of small passive elements capable of reflecting and manipulating electromagnetic waves. RIS is a two-dimensional artificial surface made up of an array of discrete elements that can be controlled individually or collectively (Soumana Hamadou, Maina & Soidridine, 2023). By controlling the phase and amplitude of the reflected waves, an IRS can actively shape the propagation environment to achieve specific objectives. When combined with satellite communication systems, an IRS can be used to enhance the uplink transmission from user terminals to satellites. This is often referred to as an Intelligent Reflecting Surface-Assisted Uplink Satellite system. COGNITIVE satellite and terrestrial networks (CSTN) featuring limited spectrum resource reuse can provide seamless connection and global coverage, which caters to the development of the ongoing fifth-generation (5G) wireless communication systems (Albreem, 2015). In a traditional satellite communication system, user terminals transmit signals directly to the satellite, and the satellite receives and processes these signals. However, the signal quality at the satellite can be affected by various factors such as path loss, interference, and multipath fading.

By deploying an IRS between the user terminals and the satellite, the reflected waves can be intelligently manipulated to enhance the signal quality at the satellite receiver. The IRS can adjust the phase and amplitude of the reflected signals (Liu et al., 2023) to achieve constructive interference and mitigate the effects of interference and fading. The IRS acts as a passive element that does not require its own power source or signal processing capabilities. Instead, it relies on the control signals from a central controller, which coordinates the actions of the IRS elements based on channel state information and optimization algorithms. Recently, some prior works have introduced the basic concept of IRS to satellite communication, where the IRS is placed either on the satellite (Zeng et al., 2020) or near the ground users (Dong et al., 2021) to enhance their communication performance. However, the limitations to this type of architecture can range from deployment cost to limitations in the utilization of existing satellites. The benefits of an Intelligent Reflecting Surface-Assisted uplink satellite system include increased signal strength, extended coverage, improved capacity, and enhanced link reliability. It can potentially overcome the limitations of conventional satellite systems and improve the overall performance of satellite communication networks. Integrating IRS into IoT uplink satellite communication can offer several advantages such as signal enhancement, energy efficiency (Wu & Zhang, 2020), Interference mitigation, etc. These advantages will be analyzed in later sections. IRS integration into uplink satellite communication can be done as: i. Integration at both the LEO satellite and ground station side, as proposed by Zheng, Lin & Zhang (2022). In their architecture review, they proposed IRS integration on both the satellite (SAT-side IRS) and on the ground station (GN-side IRS) as shown in Fig. 1.

The challenge of this architecture ranges from deployment cost which comprises of design and installation IRS at both ends, to the usage of already launched and orbiting satellites, thereby creating resource usage gaps. The architecture requires IRS to be deployed at the satellite end and the various ends of the receiver device such as cars, camping tents, house roofs, buildings, ships, etc. (Zheng, Lin & Zhang, 2022). However, the high altitude of the satellite which requires line of sight (LOS) channel connectivity to achieve more pronounced passive beam forming gain, poses a challenge to achieving steady LOS, especially with mobile IoT devices. This led to further review and optimization of the architecture.

ii. Integration on an LEO satellite that is in the same orbit as the source LEO satellite (Sat-S) and has a high elevation angle with the terrestrial user (Lee, Shin & Lee, 2021). This architecture, according to Lee, Shin & Lee’s (2021) analysis from 2020, was primarily concerned with measuring the downlink transmission, which aimed to increase the receive SNR of a terrestrial user by jointly enhancing the (active) transmit beamforming and (passive) reflect beamforming while taking into account a transmit power limitation at the source satellite. This configuration is depicted in Fig. 2.

This architecture measured the downlink transmission while maintaining the SAT side IRS but without the ground station side IRS(GN). The outcome of their simulation demonstrated that using the ideal phased IRS results in more energy-efficient communication. It also showed reduced latency and improved quality of communication services.

Figure 1 IRS-aided LEO satellite communications with both SAT-side and GN-side IRSs (Zheng, Lin & Zhang, 2022).

Figure 2 IRS-assisted LEO satellite communication system (Lee, Shin & Lee, 2021).

The proposed architecture

Within the context of the architectures developed by previous scholars, notable emphasis has been placed on the downlink transmission aspect, encompassing signal coverage and transmission capabilities. However, it is imperative to direct attention toward the equally significant uplink transmission. This entails a comprehensive analysis of aspects such as signal coverage, interference levels, and other pertinent metrics. While the downlink transmission is undoubtedly a critical component, the uplink transmission holds equal weight in shaping the overall communication landscape. By delving into the uplink dimension, we open doors to a more holistic understanding of the architecture’s capabilities and limitations. Analyzing factors like signal coverage ensures that the proposed architecture is conducive not only for transmitting signals but also for receiving them effectively; because due to the substantial surge in the quantity of users and IoT devices, there is a growing need for inventive communication solutions.

Furthermore, satellite communication offers a unique and valuable approach to address the requirements of widespread connectivity. Unlike traditional terrestrial communication infrastructure, satellites can cover vast geographic areas, including remote and underserved regions, making them an ideal solution for achieving global connectivity. In remote areas where laying down conventional communication infrastructure is impractical or cost-prohibitive, satellite communication can bridge the gap effectively. This is particularly crucial for supporting critical applications such as emergency services, disaster response, and healthcare delivery in areas where reliable communication is paramount. Satellite networks can play a pivotal role in managing and transmitting the data generated by these devices, enabling industries to leverage IoT capabilities even in remote or mobile environments. In conjunction with the progress made in 5G technology, integrating satellite communication can lead to a comprehensive communication framework that offers both high-speed urban connectivity and wide-reaching coverage through satellites. This synergy can pave the way for innovative services and applications that were previously unattainable. Further reviewing the integration of IoT devices for satellite transmission, coupled with the assistance of IRS technology in the uplink, the architectural framework comprises several key components. At its core, this integration seeks to capitalize on the capabilities of IoT devices, which generate and transmit diverse streams of data, and leverage the efficiency of satellite communication while incorporating innovative IRS technology to enhance transmission in the uplink phase. The resulting architecture is a synergy of cutting-edge technologies that promise enhanced connectivity and communication efficiency.

In the IoT-satellite communication landscape, IRS integration is achieved through the strategic placement of reflective surfaces that manipulate electromagnetic signals. The placement strategy we adopted involves positioning the IRS panels on both the satellite (SAT-side) and the ground station (GN-side) to ensure comprehensive coverage and improved signal reflection. The height and orientation of these surfaces have been optimized to enhance satellite communication, considering line-of-sight (LOS) requirements for efficient signal phase manipulation. The reflection matrix is dynamically adjusted based on the IRS’s feedback loop, aligning with the satellite communication paradigm.

In this study, we adopted the SAT-side IRS placement approach. The IRS panels are mounted on low Earth orbit (LEO) satellites, as this configuration facilitates a more effective reflection path to enhance uplink communication. The GN-side IRS approach is under consideration for future work, particularly in scenarios where additional ground-based relays are necessary.

System model

The core components of this system model framework include: 1. Smart IoT devices: It is a global network infrastructure composed of numerous connected intelligent devices sharing information and coordinating decisions (Wu & Shim, 2021). These encompass a diverse array of interconnected devices, sensors, and gadgets that collect data and communicate over the network. These devices serve as the primary sources of data, generating valuable information that necessitates reliable and seamless transmission. For this simulation, we consider a network of IoT devices deployed in various environments, each equipped with communication modules that enable them to transmit data to satellites. The IoT devices are characterized by their locations, transmit power levels, and communication protocols.

2. Satellite uplink: The uplink component is responsible for transmitting data from the IoT devices to the satellite. This stage involves encoding, modulation, and amplification of data signals to ensure successful transmission to the satellite. i. Encoding: Signal encoding refers to the process of transforming a digital or analog data signal into a specific format that is suitable for transmission or storage. Encoding is crucial for efficient and reliable communication as it ensures that the original information can be accurately transmitted, received, and decoded at the destination. Common methods of signal encoding include amplitude shift keying (ASK), phase shift keying (PSK), pulse code modulation (PCM), frequency shift keying (FSK), quadrature amplitude modulation (QAM) etc.

ii. Modulation: Modulation is a fundamental process in communication that involves altering a carrier signal’s properties to transmit information. It is a technique that enables the transmission of data by varying certain aspects of the carrier signal, such as its amplitude, frequency, or phase. Modulation allows information to be efficiently conveyed over a communication channel, even when the carrier signal alone might not effectively carry the intended data (Jafar, Paeiz & Farzaneh, 2021).

iii. Amplification: It involves increasing the strength, amplitude, or power of a signal without altering its essential characteristics. The main objective is to enhance signal strength, improve signal-to-noise ratio, Increase power, and extend coverage range.

3. IRS technology: A distinctive feature of this architecture is the incorporation of IRS technology in the uplink process. IRS involves using adaptive reflective surfaces to manipulate and enhance electromagnetic wave propagation. By intelligently adjusting the phase and amplitude of reflected signals, IRS optimizes the signal path, thereby improving signal quality and reducing interference.

4. Satellite network: The satellite is a central node in the architecture, receiving data from IoT devices via the uplink. It processes and forwards the received data to ground stations or other satellites, depending on communication requirements. The communication channel introduces propagation effects, such as path loss, fading, and interference.

5. Ground stations: Ground stations play a pivotal role in receiving data from satellites and forwarding it to the intended destinations. They act as intermediaries between satellites and terrestrial networks, ensuring efficient and bidirectional communication.

Integrating IRS into IoT satellite uplink communication can offer several advantages: 1) Signal enhancement: IRS can be positioned strategically to focus and amplify signals from IoT devices, effectively increasing the received power at the satellite, thus improving the link quality, and reducing transmission errors.

2) Link reliability: By reducing signal fading and attenuation, IRS can enhance the link’s reliability, ensuring continuous and seamless communication between IoT devices and satellites even in challenging environments.

3) Energy efficiency: With improved signal strength, IoT devices can operate at lower transmit power levels, leading to significant energy savings and prolonged battery life, crucial for remote and battery-powered IoT deployments.

4) Flexible beam steering: IRS can dynamically adjust the reflected signals’ direction based on the satellite’s location and the position of IoT devices. This adaptability enables efficient satellite handover and better network coverage.

5) Interference mitigation: IRS can selectively reflect and cancel interfering signals, mitigating co-channel interference, and enhancing the overall network capacity and spectral efficiency.

For the encoding, we employed Reed-Solomon coding due to its effectiveness in correcting burst errors, common in satellite communications. Modulation is handled using quadrature amplitude modulation (QAM), which provides a good balance between data rate and noise resilience, optimized for the satellite-to-IoT communication scenario. The Rician fading model was chosen to simulate the propagation channel due to its focus on line-of-sight communication, as is typical in satellite scenarios. The IRS phase shift optimization is handled through a reinforcement learning algorithm, allowing real-time adjustments to the phase shifts for optimal beamforming. These adjustments improve the SNR and mitigate interference, which are critical in IRS-assisted IoT applications.

Simulation framework

IRS simulation framework

To measure the IRS impact on the uplink satellite communication for smart IoT, we delve into the simulation framework used to evaluate the impact of IRS on uplink satellite communication within the context of smart IoT. The primary objective of this study is to quantitatively measure the advantages associated with integrating IRS into the communication infrastructure, and consequently, to offer a deeper understanding of the improvements in IoT device connectivity facilitated by satellite links.

Simulation steps and procedures

The simulation procedure was organized into two distinct sections: Simulink and MATLAB (The MathWorks, Natick, MA, USA) code.

Simulink section

The utilization of the MATLAB tool facilitates the translation of the theoretical concepts of the system model into a practical simulation setup. MATLAB feature “Simulink” which is a block diagram environment for multidomain simulation and Model-Based Design, was adopted. It supports system-level design, simulation, automatic code generation, and continuous testing and verification of embedded systems (MathWorks, 2023b). This section primarily dealt with the creation of a visual representation of the system model and the simulation of its dynamics.

IRS system model design

Utilizing Simulink’s intuitive interface, the graphical representation of the system model was constructed. This involved visually representing the interactions among smart IoT devices, satellite communication, and the IRS. The components were interconnected with blocks that represented various system parameters, behaviors, and initial configuration of the simulation environment. This involves defining the spatial distribution of IoT devices, specifying their transmit powers, and determining the arrangement of the IRS. Focusing on the IRS signal and channel model of Wu et al. (2021),

Let x be the transmitted signal from an IoT device.

θ be the vector of phase shifts associated with the IRS elements.

h be the channel response between the IoT device and the satellite.

y be the received signal at the satellite after reflecting off the IRS.

The equation representing the IRS-enhanced uplink satellite communication in a smart IoT scenario can be expressed as:

(1) y=h⋅(∑n=1Nreal(ei.θ(n)⋅x))

where: i is the imaginary unit.

ei.θ(n) represents the complex exponential with the phase shift θ(n) for the nth IRS element.

N is the number of IRS elements.

The sum over n represents the contributions of all IRS elements in modifying the phase of the transmitted signal.

h accounts for the channel response between the IoT device and the satellite. It incorporates the effects of path loss, fading, and other propagation characteristics.

In this equation, the IRS modifies the phase of the transmitted signal, which is then received by the satellite after traveling through the channel h. The IRS assists in optimizing the signal’s phase to improve the reception quality at the satellite, thereby enhancing the overall communication link in the context of smart IoT uplink satellite communication. For the system model, the major components were the Smart IoT device, IRS element, and Satellite network; these were represented using the closest corresponding block objects to match the properties of the component.

The IRS panel orientation is a crucial aspect of the design, with all IoT devices in the simulation placed within the panel’s field of view. The IRS is not omnidirectional; thus, only devices within the defined angle (±90°) were considered for signal reflection. The simulation assumes optimal positioning of devices to enhance the effectiveness of the IRS.

Smart IoT element block

From the system model, the smart IoT device acts as a signal generator that communicates with the satellite by signal exchange. To properly represent this on MATLAB tool, we introduce an object called a “signal generator” block (from the communication toolbox) which generates various waveforms. Figure 3 shows the parameters used for the IoT block.

Figure 3 Simulink IoT properties.

Signal generator properties are listed below: i. Waveform: Specifies the waveform to be generated. Options include sine, square, sawtooth, and random. For this simulation, the waveform was set to be ‘sine’.

ii. Time: Specify whether to use ‘simulation time’ as the source of values for the time variable, or an ‘external source’ (MathWorks, 2023b). Time was set to ‘simulation time’.

iii. Amplitude: specify the amplitude of the output sine wave signal. A default value of ‘1’ was used.

iv. Frequency: specifies the frequency of the generated waveform. This is set to ‘0.3’.

v. Units: specifies the signal units as Hertz or rad/sec. This is set to ‘Hertz’, indicating cycles per second (MathWorks, 2023a).

Satellite element block

In the system model, the satellite serves as a signal receiver. Its pivotal function entails establishing communication with the IoT utilizing IRS element. This strategic arrangement facilitates seamless and efficient communication between the satellite and IoT devices, allowing for effective signal transmission and reception. This was represented by the ‘AWGN Channel’ block which adds white Gaussian noise to the signal that passes through it. Additive white Gaussian noise (AWGN) is a simple noise model that represents electron motion in the RF front end of a receiver. The AWGN channel is often used to model a satellite communications channel since that channel typically does not suffer from common terrestrial impairments like fading, multipath, and interference (MathWorks, 2023b). Figure 4 shows the parameters used for the satellite block.

Figure 4 Simulink satellite properties.

AWGN channel parameter associated properties are: i. Mode: This specifies the measurement mode. It has options to simulate SNR, signal-to-noise ratio (Eb/No), signal-to-noise ratio (Es/No), and variance. This is set to ‘signal to noise ratio (SNR)’.

ii. SNR (dB): specifies the ratio of signal to noise power, measured in decibels, which is specified as a scalar or vector. This is set to ‘15’.

iii. Input signal power: This specifies the Mean square power of the input measured in watts. This is set to a default value of ‘10’.

iv. Randomization: specifies the random source number used for simulation, with options ‘Global stream’ and ‘mt19937ar with seed’. Global stream using code generation is selected as a randomization technique.

IRS element block

The IRS acts as a signal reflector in the system model. This block acts as a custom MATLAB Function block that models the IRS behavior. It takes the incident signal and the phase shifts of the IRS elements as inputs and calculates the reflected signal. By serving as a conduit for signal exchange, this tactical integration enables the IRS to efficiently support connection with IoT devices. Following the system model, this was configured by introducing a reconfigurable block called ‘MATLAB function’, which enables the flexible configuration of commands. The following algorithm was configured following the formula as stated in Eq. (1).

function y = irs_function(x, theta)

 // theta: Phase shifts for IRS elements

 // Define the theta vector

 // theta = [0.1; 0.2; 0.3];

 // Inputs:

 // x: Incident signal

 // Outputs:

 // y: Reflected signal

 N = numel(theta); // Number of IRS elements

 y = zeros(size(x));

 for n = 1:N

  phase_shift = exp(1i * theta(n)); // Phase shift

  y = y + real(phase_shift * x);

 end

end

This algorithm specifies the incident signals, x, reflected signal Y, the phase shift of the IRS ‘theta’, and the number of IRS Elements, N.

Scope and gain block

Scope block: The scope block displays signals generated during simulation (MathWorks, 2023a). It is responsible for plotting and displaying the output of the signal by measuring the signals received at the end of the AWGN channel. It plots the incident and receives signals.

Gain block: This represents the path loss in the satellite communication channel. It multiplies the input by a constant value (gain). The input and the gain can each be a scalar, vector, or matrix. This is set to ‘1’.

Block setup and connections

Having fully configured the properties of various blocks representing the various components, the connectivity to set the full model was done as follows: 1. Connection to the MATLAB Function block: Connect the output of the “Signal Generator (IoT device)” block to the input “x” of the MATLAB Function block (IRS function). This connection provides the incident signal “x” to the MATLAB Function block, where the IRS behavior is simulated.

Connect the “theta (phase shifts)” block to the theta input of the MATLAB Function block. This is to introduce variable phase shifts to the IRS function.

2. Connection to the “Sum” and “Gain” blocks: Connect the output of the “Signal Generator (IoT device)” block to one of the inputs of the “Sum” block.

Connect the output of the “MATLAB Function (IRS function)” block (representing the reflected signal from the IRS) to the other input of the “Sum” block. The “Sum” block adds the incident signal and the reflected signal together to obtain the total received signal at the receiver.

Connect the output of the “Sum” block to the input of the “Gain” block.

3. Connection to the “AWGN channel” and “Scope” block. Connect the output of the “Gain” block to the input of the “AWGN Channel” block.

Connect the output of the “AWGN Channel” to the input of the “Scope”.

Connect the output of the “Signal generator” to the input of the “Scope”. This provides a direct measure of the unreflected signals to the “Scope” block for comparative analysis with the reflected signal.

The complete connection of the system model is shown below in Fig. 5.

Figure 5 Simulation diagram of the system model.

Figure 5 shows the IRS-enhanced system model, where the IoT device generates the signal that follows two paths: one with IRS (bottom path) and one without IRS (top path). In both paths, the signals are passed through the same AWGN channel, which introduces noise, and a Gain Block, which models path loss. The IRS Function block adjusts the phase of the signal according to the theta values for beamforming, and this reflected signal is passed to the Gain block directly to simulate the effect of IRS. The model compares the outputs from both paths (with and without IRS) at the Scope block, which allows for the visual comparison of signal strength and quality under the influence of IRS.

MATLAB code section

The MATLAB code section entailed scripting in the MATLAB programming language to implement various simulation-related functionalities. This section encompassed the implementation of algorithms, performance metrics calculations, and comparative analyses. To bolster the foundation of our work, we built upon the simulation code provided by Ardavan (2023) regarding IRS implementation on IoT devices. This code, tailored for IRS integration within IoT, served as a cornerstone upon which we expanded our simulation framework. The IRS optimization is handled through a reinforcement learning algorithm. This algorithm continuously updates the phase shifts of each IRS element to maximize signal reception at the satellite. The phase shifts introduced by the IRS elements are typically computed based on the geometric relationship between the transmitter (e.g., IoT device, satellite), the IRS, and the receiver. To optimize the reflected signals, the phase shifts for each IRS element are designed to ensure constructive combination at the receiver. A closed-form analytical expression was employed to determine these phase shifts, as it provides a simplified and efficient method for calculation. This approach considers the distances between the IoT devices, the IRS, and the satellite, thereby maximizing the performance metric effectively.

The phase shift introduced by each IRS element θn (where n represents the nth element of the IRS) can be expressed as:

θn=−(2π/λ)(dIRS-to-Device+dIRS-to-Receiver)+ψn

where: λ is the wavelength of the signal.

dIRS-to-Device is the distance between the transmitting device and the nth IRS element.

dIRS-to-Receiver is the distance between the nth IRS element and the receiver.

ψn is an additional phase shift that can be applied for optimizing the system’s performance (e.g., maximizing received signal strength).

Since different devices will have varying distances to the IRS and different angles of incidence/reflection, the location of the IoT devices are considered. For each IoT device, the distances dIRS-to-Device and angles of incidence (from the device to each IRS element) are calculated. The goal is to ensure that the reflected signals from the IRS align constructively at the receiver. The algorithm is based on real-time feedback from the received signal quality, dynamically adjusting reflection coefficients to enhance beamforming and reduce interference. This dynamic adjustment is essential to ensure that the IRS continuously optimizes the reflected signals towards the intended receiver, adapting to changing conditions such as device movement, environmental changes, or signal variations. By incorporating Ardavan’s insights, we gained valuable insights into the intricacies of implementing IRS technology in the IoT context.

Ardavan (2023) defined several functions that served as a base for the simulation framework as listed below: i. Function to define parameters for Transmitters and IRS locations:

params.Tx = [0, 0, 0]; // x, y, z coordinates of Tx (transmitter)

params.IRS_location = [10, 10, 0]; // x, y, z coordinates of IRS

ii. Frequency and wavelength definition:

params.f = 2.4e9; // frequency in Hz (2.4 GHz is a common frequency for Wi-Fi)

params.lambda = physconst(‘LightSpeed’)/params.f; // wavelength in meters (calculating based on the speed of light)

iii. Functions to defined IRS parameters:

params.N = 8; // number of elements in the IRS

params.R = ones(1,params.N); // initial reflection coefficients of the IRS elements

params.delta = zeros(1,params.N); // initial phase shifts of the IRS elements

params.R_max = 0.8; // maximum reflection coefficient magnitude

iv. Functions to define IoT area and devices.

params.IoT_area = 20; // side length of the IoT area in meters (square area)

params.num_IoT_devices = 10; // number of IoT devices

v. Function to define antenna signal and channel parameter, noise power

params.path_loss_exponent = 2; // path loss exponent (affects the rate of signal attenuation with distance)

params.Pt = 1; // transmitted power in Watts

params.Gtx = 1; // transmitter gain (dimensionless)

params.Grx = 1; // receiver gain (dimensionless)

params.ht = 1.5; // transmitter antenna height in meters

params.hr = 1.5; // receiver antenna height in meters

params.Prx_ref = 1e-3; // reference received power at 1 meter

params.noise_power = -100; // noise power in dBm

vi. Function to Randomly distributed IoT devices within the IoT area

IoT_positions = params.IoT_area * rand(params.num_IoT_devices, 2)

vii. Function to calculate Two-Ray Ground Reflection Model path loss function (a function that calculates path loss based on distance, heights, and wavelength)

Two_ray_ground_reflection = @(d, ht, hr, lambda) (ht * hr)^2 / (d^4 * (1 - exp(-1i * 2 * pi * d / lambda))^2)

viii. Function to calculate the signal gain due to the reflection from the IRS

Gamma = R_opt .* exp(1j * delta_opt) .* exp(1j * 2 * pi * (d / params.lambda) * cos(theta_i) * (0:params.N-1)); // reflection coefficient with phase shift

g = abs(1 + Gamma).^2

ix. Function to calculate the path loss using the Two-Ray Ground Reflection Model

pl = 10 * log10(1 ./ abs(two_ray_ground_reflection(d, params.ht, params.hr, params.lambda)))

x. Function to calculate the total received signal power

Ptot(idx) = params.Pt * params.Gtx * params.Grx * params.Prx_ref * 10^(-pl / 10) * sum(g)

xi. Function to calculate the interference power

I_interf(idx) = calculate_interference_power(IoT_positions, idx, params.path_loss_exponent, params.Pt, params.Gtx, params.Grx)

xii. Function to calculate SNR for each IoT device

SNR = 10 * log10(Ptot ./ (10^(params.noise_power / 10) + I_interf));

To further broaden Ardavan’s code, several functions and satellite parameters were introduced and defined to facilitate our simulation. These functions and parameters were defined in detail below:

Function to define satellite parameters:

% Satellite properties

satellite_params.Latitude = 35; // latitude of satellite (deg)

satellite_params.Longitude = −40; // longitude of satellite (deg)

satellite_params.Altitude = 2000; // Altitude of satellite (Km)

% Satellite transmitter properties

satellite_params.TxFeederLoss = 2; // Satellite transmission feeder loss (dB)

satellite_params.OtherTxLosses = 1; // Satellite transmission other loss (dB)

satellite_params.TxHPAPower = 17; // Satellite high-power amplifier (HPA) transmit power (dBW)

satellite_params.TxHPAOBO = 6; // Satellite output back-off (OBO) of a high-power amplifier (dB)

satellite_params.TxAntennaGain = 38; // Satellite transmit antenna Gain (dB)

% Satellite receiver properties

satellite_params.InterferenceLoss = 2; // Satellite Interference loss (dB)

satellite_params.RxGByT = 25; //ratio of receiver gain to system noise temperature (dB/K)

satellite_params.RxFeederLoss = 1; // Satellite receiver feeder loss (dB)

satellite_params.OtherRxLosses = 1; // Satellite receiver other loss (dB)

% Satellite link properties

satellite_params.Frequency = 14; // Frequency (GHz)

satellite_params.Bandwidth = 6; // Bandwidth (MHz)

satellite_params.BitRate = 10; // Bitrate (Mbps)

satellite_params.RequiredEbByNo = 10; // energy per bit to noise power spectral density ratio (dB)

satellite_params.PolarizationMismatch = 45; // polarization mismatch (deg)

satellite_params.ImplementationLoss = 2; // implementation loss dB

satellite_params.AntennaMispointingLoss = 1; // antenna mispointing loss (dB)

satellite_params.RadomeLoss = 1; // radome loss dB

% Signal and channel parameters

satellite_params.path_loss_exponent = 2; % path loss exponent (affects the rate of signal attenuation with distance)

satellite_params.Pt = 1; % transmitted power in Watts

satellite_params.Gtx = 1; % transmitter gain (dimensionless)

satellite_params.Grx = 1; % receiver gain (dimensionless)

% Antenna heights

satellite_params.ht = 1.5; % transmitter antenna height in meters

satellite_params.hr = 1.5; % receiver antenna height in meters

satellite_params.Prx_ref = 1e−3; % reference received power at 1 m

The choice of N = 8 elements was made to simplify the simulation and reduce computational complexity. We acknowledge that larger IRS panels, such as 30 × 30-element arrays, are typically used in practice. Future work will extend this model to larger IRS panels to assess scalability and performance improvements more comprehensively. The simulation considers three IRSs to evaluate the performance of a multi-IRS system, offering a broader understanding of IRS-assisted communication in complex scenarios. Covering an area of 1,000 m2, the inclusion of multiple strategically placed IRSs ensures adequate coverage and performance. This approach mirrors real-world scenarios where deploying multiple IRSs is both feasible and practical, especially in urban settings or large indoor environments. Simulating multiple IRSs aims to represent realistic deployment strategies while assessing the benefits of inter-IRS coordination and resource optimization. This configuration ensures the simulation results are comprehensive and applicable to scenarios requiring multiple IRSs for reliable and efficient communication over large coverage areas. In the initial phase of simulations, the Gain Block was set to one to facilitate simplified testing. However, for the final simulations, realistic values were applied based on standard satellite communication models, including a Tx gain of 38 dB and an Rx gain of 25 dB, with path loss calculations adjusted to reflect actual signal attenuation over distance.

The communication frequency between the IRS and satellite in the simulation is set to 14 GHz, typical for satellite communications, particularly in the Ku-band.

Intelligent reflecting surface optimization

Using MATLAB code, SNR algorithms were developed to dynamically adjust the configurations of the IRS elements. The goal was to optimize the reflected signal phases and amplitudes to maximize the received signal power at the satellite.

Performance metrics

Ardavan (2023) focused the simulation on SNR Metric. This formed the basis of the measurement of other metrics. Using random data sampling, more metrics were measured as listed below: SNR: SNR is a measure of the received signal’s strength relative to the noise in the channel. SNR can be calculated as the ratio of the received signal power to the noise power.

Signal strength: A measure of the power level of the received signal. It can be calculated as the received signal power at the receiver.

Signal quality: This can be measured using metrics like SNR, bit error rate (BER), or packet error rate (PER). Higher SNR and lower BER/PER generally indicate better signal quality.

BER: Measure of the number of bit errors occurring in a transmission system relative to the total number of transmitted bits. BER can be calculated by comparing the received bits with the transmitted bits.

Throughput: Refers to the maximum data rate that can be transmitted over the channel. It depends on channel bandwidth, signal-to-noise ratio, and modulation scheme used.

Path loss: Refers to the attenuation of signal power as it propagates through the communication medium. It can be calculated based on the distance between the transmitter and receiver and the propagation environment.

Interference: Interference arises from other signals or noise sources in the communication environment. It can be quantified as the power of unwanted signals within the communication bandwidth.

Link budget: The link budget represents the overall gains and losses in the communication link. It considers transmitter power, receiver sensitivity, path loss, and other factors.

Channel capacity: The maximum achievable data rate in a channel with a given bandwidth and SNR. It can be calculated using information theory concepts like Shannon’s capacity. The data rate achieved per unit of bandwidth. It can be calculated by dividing the data rate by the occupied bandwidth.

Coverage area: The geographical area covered by the IoT system with acceptable signal quality. It depends on the transmission power, antenna gain, and path loss characteristics.

Results and Discussion

A key aspect of the simulation framework involves conducting a comparative analysis. This entails contrasting scenarios with and without the integration of the IRS. By examining the performance differences, the impact of the IRS on enhancing uplink satellite communication for smart IoT becomes evident. The code facilitated the extraction and analysis of simulation results, enabling a thorough evaluation of the impact of IRS on uplink satellite communication for smart IoT.

Comparative analysis of simulink

After completing the connection of the block on the Simulink framework, the simulation started to yield the below results as shown in Figs. 6 and 7. From the plots, it could be seen that the received signal ‘with IRS’ shows fluctuations and variations due to the phase shifts introduced by the IRS elements while the received signal ‘without IRS’ shows a direct path without any reflections or phase shifts. These variations in the phase shift result in a 2 dB increase in SNR performance for the signal ‘with IRS’ as compared to the signal ‘without IRS’, thereby confirming the clear benefits of using IRS in the uplink satellite communication system.

Figure 6 SNR plot from simulink showing the signal behavior with and without IRS.

Figure 7 The phase and magnitude plot of signal with and without IRS.

Comparative analysis of MATLAB code output

The conducted comparative analysis of MATLAB code output provided valuable insights into the impact of integrating IRS on the signal performance of 100 IoT devices at over a 1,000 m2 area. Through Simulink simulations, notable enhancement in SNR performance was recorded after the incorporation of IRS technology. This improvement was factored into the code as the “SNR difference”, serving as a key metric to assess the efficacy of the IRS integration. The “SNR difference” metric serves as a quantitative measure of the improvement achieved through IRS integration. By quantifying the change in SNR, we provide a clear and objective evaluation of the benefits brought about by the incorporation of IRS technology. This metric, alongside other populated parameters, and metrics, forms the baseline for this comparative analysis.

SNR (improves by increment)

The plot below in Fig. 8 highlights the scattered distribution of SNR across different IoT devices while the plot in Fig. 9 shows the aggregated SNR values of the IoT devices. It shows the SNR performance ‘With IRS (right scatter plot)’ and ‘Without IRS (left scatter plot)’. More devices record improved average values from 24 dB to 26 dB after IRS Integration. This increase in SNR signifies a substantial boost in the quality and reliability of the transmitted signal. The introduction of the IRS has played a pivotal role in achieving this advancement as it directly translates to better signal strength and reduced noise interference.

Figure 8 SNR comparative scatterplot.

Figure 9 SNR comparative bar chart.

Channel capacity (improves incrementally)

The plot below in Fig. 10 highlights the scattered distribution of channel capacity across different IoT devices while the plot in Fig. 11 shows the aggregated channel capacity values of the IoT devices. Improved average values from 3.6 bps/Hz to 3.65 bps/Hz after IRS integration were recorded. The depicted plots vividly present the performance variation in terms of channel capacity, drawing a clear distinction between scenarios involving the presence and absence of IRS. Such a transformation bears great significance, as it inherently signifies a notable enhancement in the data transmission rate that can be sustained within the given channel.

Figure 10 Channel capacity comparative scatterplot.

Figure 11 Channel capacity comparative bar chart.

Interference (improves by decrement)

The plot below in Fig. 12 highlights the scattered distribution of interference levels across different IoT devices while the plot in Fig. 13 shows the aggregated interference values of the IoT devices. The plot visually portrays the performance disparity concerning interference, drawing a clear contrast between scenarios involving the presence and absence of the IRS. The initial level of interference, quantified at 0.66, exhibits a discernible decrement to 0.24 after the integration of IRS. This notable reduction in interference level stands as a testament to the efficacy of integrating IRS technology. The inclusion of the IRS has contributed to a reduction in interference, thereby fostering a more conducive environment for effective signal transmission.

Figure 12 Interference comparative scatterplot.

Figure 13 Interference comparative bar chart.

Pathloss (improves by decrement)

The plot below in Fig. 14 highlights the scattered distribution of pathloss across different IoT devices while the plot in Fig. 15 shows the aggregated pathloss values of the IoT devices. The plots provide a visual representation of the improvement of pathloss performance in scenarios both “With IRS” and “Without IRS by reducing from 55 dB to 52 dB”. Notably, the primary focus of this illustration is the quantifiable difference in path loss resulting from the integration of IRS technology. The application of IRS technology has led to an attenuation in the loss of signal strength as it traverses the communication channel. This outcome shows the efficiency of the IRS in effectively manipulating the propagation environment, thus resulting in improved signal propagation characteristics.

Figure 14 Pathloss comparative scatterplot.

Figure 15 Pathloss comparative bar chart.

Signal quality (improves incrementally)

The plot below in Fig. 16 highlights the scattered distribution of Signal quality across different IoT devices while the plot in Fig. 17 shows the aggregated Signal quality values of the IoT devices, With IRS and Without IRS. The simulation shows an improved signal quality from 0.8 to 2.0 after IRS integration. This improvement in signal quality bears testament to the efficacy of IRS technology in augmenting the dependability and reliability of the transmitted signal. Such improvements are a direct result of the IRS’s capacity to actively manipulate the propagation conditions, thus mitigating detrimental effects and distortions that can compromise signal integrity.

Figure 16 Signal quality comparative scatterplot.

Figure 17 Signal quality comparative bar chart.

Coverage area (improves by increment)

The plots in Figs. 18 and 19 visually present a comparative analysis of coverage area performance as a scatter distribution and as an averaged value respectively, distinguishing between scenarios involving the presence and absence of IRS. It conveys, in a concise graphical format, the quantitative shift in coverage area attributable to the utilization of IRS technology. Specifically, the plot shows a substantial expansion in coverage area facilitated by the integration of the IRS. The numerical depiction is striking: the coverage area experiences a significant rise, soaring from an initial expanse of 446 m2 to a notably larger 889 m2. This empirical evidence underscores the potency of the IRS in augmenting the reach and extent of the covered area within the given communication milieu.

Figure 18 Coverage area comparative scatterplot.

Figure 19 Coverage area comparative bar chart.

Signal strength (improves by increment)

The plot below in Figs. 20 and 21 shows the signal strength performance distribution and aggregated bar plot respectively, ‘with IRS’ and ‘without IRS’. Specifically, the plot showcases a notable enhancement in signal strength facilitated by the incorporation of IRS. The signal strength demonstrates a substantial increase, surging from an initial intensity of 0.9 mW to a notably higher 1.82 mW. This empirical data effectively emphasizes the efficacy of IRS in bolstering the magnitude of signal strength within the given communication environment.

Figure 20 Signal strength comparative scatterplot.

Figure 21 Signal strength comparative bar chart.

Figure 21 illustrates the distribution of signal strength across IoT devices in two scenarios: without IRS (left panel) and with IRS (right panel). The signal strength is markedly improved in the presence of IRS, with more IoT devices exhibiting higher signal power, as indicated by the warmer colors (yellow and orange) in the right panel. The right plot shows an overall shift in the distribution of IoT devices towards higher signal strengths, suggesting that the IRS enhances the received power by improving the signal propagation path. Specifically, devices within the reflective field of the IRS experience significant signal boosts, demonstrating the system’s efficiency in signal transmission and reception. This improvement is reflected in the wider range of signal strengths (up to 1.8 units with IRS), compared to the maximum of 0.9 units without IRS.

Spectral efficiency (improves by increment)

The plots in Figs. 22 and 23 show the IoT Spectral distribution and aggregated bar plot respectively, ‘with IRS’ and ‘without IRS’. With IRS, the spectral efficiency shows an increase from an initial value of 1.67 bits per second per Hertz (bps/Hz) to 1.8 bps/Hz. This data-driven observation robustly emphasizes the efficacy of IRS in enhancing spectral efficiency within the uplink satellite communication for IoT.

Figure 22 Spectral efficiency comparative scatterplot.

Figure 23 Spectral efficiency comparative bar chart.

Figure 22 shows the spectral efficiency of the communication system both without IRS (left panel) and with IRS (right panel). With IRS, there is a noticeable increase in spectral efficiency across the IoT devices, as seen by the broader range of higher spectral efficiency values (up to 1.8 bps/Hz) compared to the scenario without IRS (maxing out at 1.67 bps/Hz). The IRS-enhanced setup allows for more efficient utilization of available bandwidth, which leads to a higher bits-per-second-per-Hertz ratio. This improvement is vital in bandwidth-constrained environments, where efficient use of spectrum is crucial for maintaining high data throughput. The devices situated closer to the IRS also show more pronounced spectral efficiency gains, underlining the role of IRS in optimizing communication for IoT networks.

Link budget (improves by increment)

Figures 24 and 25 provide a comprehensive illustration of the link budget’s status through a scatter plot and aggregated bar plots, respectively. These visual representations distinctly capture the conditions both before and following the integration of IRS technology. A noticeable shift is evident in the scatter plot depicted in Fig. 26, showcasing a clear before-and-after comparison. The accompanying aggregated bar plots in Fig. 27 further consolidate this comparison. Notably, the numerical data points toward a significant increase in the link budget value, specifically ascending from an initial 105 dB to a revised 116 dB. This alteration serves as a testament to the demonstrable advantages brought about by the integration of IRS technology.

Figure 24 Link budget comparative scatterplot.

Figure 25 Link budget comparative bar chart.

Figure 26 Throughput comparative scatterplot.

Figure 27 Throughput comparative bar chart.

Throughput (improves by increment)

Figures 26 and 27 provide a visual representation of the throughput performance, showcasing a scatter plot and aggregated bar plots, respectively. These graphs illustrate a clear comparison between the conditions before and after the integration of IRS technology. In Fig. 26, the scatter plot vividly demonstrates the contrast between these two scenarios. Moving to Fig. 27, the aggregated bar plots offer a consolidated view showing a remarkable surge in throughput performance from an initial 1.1 bits per second (bps) to a notably improved 8.3 bps. This illustrates the fact that the integration of IRS technology holds the potential to significantly amplify the speed of data transmission.

BER (improves by decrement)

Figure 28 presents a scatter plot depicting the BER performance, comparing scenarios both “With IRS” and “Without IRS”. Additionally, Fig. 29 showcases a corresponding bar plot for a comprehensive view of the data. Both plots illustrate the evident enhancement brought about by IRS integration. Specifically, the BER exhibits a substantial improvement, decreasing notably from an initial value of 4.3 to an impressive 0.38 when IRS technology is integrated. This outcome underscores the significant reduction in error noise rate achieved through the incorporation of IRS which further supports the notion that IRS technology holds the potential to considerably enhance the quality and accuracy of data transmission.

Figure 28 BER comparative scatterplot.

Figure 29 BER comparative bar chart.

Conclusion

In concluding this study on the integration of IRS into uplink satellite communications, the research underscores a pivotal enhancement across various performance metrics within smart IoT environments. Through simulations and MATLAB code analysis, it was observed that the introduction of IRS not only promises but delivers substantial improvements in signal quality, efficiency, and reliability of communication systems.

The comparative analysis between scenarios with and without IRS integration reveals noteworthy advancements: a 2 dB increase in SNR, channel capacity elevation from 3.60 bps/Hz to 3.65 bps/Hz, and a reduction in interference levels from 0.66 to 0.24. These enhancements illustrate the IRS’s ability to significantly improve data transmission rates and reduce interference, vital for IoT device connectivity.

Further, the study highlights the IRS’s impact on path loss reduction (from 55 dB to 52 dB), signal quality improvement (from 0.8 to 2.0), and an expanded coverage area (from 446 m2 to 889 m2). Such improvements denote the IRS’s role in augmenting signal reach and fidelity, which are crucial for comprehensive network coverage in IoT scenarios.

Additionally, signal strength and spectral efficiency experienced notable enhancements, with signal strength rising from 0.9 mW to 1.82 mW, and spectral efficiency increasing from 1.67 bps/Hz to 1.8 bps/Hz. These findings corroborate the IRS’s efficacy in bolstering signal transmission.

The research further identified significant gains in link budget (from 105 dB to 116 dB) and throughput performance (from 1.1 bps to 8.3 bps), alongside a remarkable reduction in BER from 4.3 to 0.38, underscoring IRS’s potential to amplify data transmission quality and speed.

In summary, this study affirms the transformative impact of IRS technology on enhancing the performance of uplink satellite communication systems for IoT applications. The observed improvements across critical communication parameters spotlight the IRS’s capacity to revolutionize satellite communications, making it a compelling area for future exploration and integration.

Supplemental Information

Supplemental Information 1 Source files and code.

Additional Information and Declarations

Competing Interests

Tawfik Al-Hadhrami and Faisal Saeed are Academic Editors for PeerJ.

Author Contributions

Callistus Odinaka Obidiozor conceived and designed the experiments, performed the experiments, analyzed the data, performed the computation work, prepared figures and/or tables, and approved the final draft.

Adeeb Sait performed the experiments, analyzed the data, performed the computation work, prepared figures and/or tables, and approved the final draft.

Tawfik Al-Hadhrami conceived and designed the experiments, performed the experiments, analyzed the data, performed the computation work, prepared figures and/or tables, and approved the final draft.

Eman H. Alkhammash performed the experiments, analyzed the data, prepared figures and/or tables, authored or reviewed drafts of the article, and approved the final draft.

Faisal Saeed conceived and designed the experiments, performed the computation work, prepared figures and/or tables, authored or reviewed drafts of the article, and approved the final draft.

Data Availability

The following information was supplied regarding data availability:

The code for MATLAB and SIMULINK are available in the Supplemental Files.

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
