# Peer review of "Intelligent reflective surfaces in 5G and beyond: optimizing uplink satellite connectivity for IoT"

_PeerJ Computer Science, doi:10.7717/peerj-cs.2726_

## Round 0.1 · original submission · Major Revisions

The reviewers have raised several concerns that need to be addressed.

Reviewer 1 ·

Basic reporting

Good.

Experimental design

GOod

Validity of the findings

RIS is applied to the uplink of satellite communications. This topic is interesting, that belongs to B5G or 6G. The main body of the whole paper is over the simulations without solid derivations.

Additional comments

None.

Reviewer 2 ·

Basic reporting

In this paper, the authors have highlighted the potential of IRS in optimising satellite connectivity for IoT applications. The intelligent reflecting surfaces are metasurfaces that enable smart radio by introducing controlled reflection. The IRS offers improved communication by optimising the signal reception and transmission. However, the use of IRS for optimising the satellite connectivity has several challenges which are not taken into account in this paper.

Experimental design

2. In the IoT-satellite communication landscape, how the IRS integration is taking place? Which IRS placement strategy is adopted. Practical things like height of IRS, its placement, orientation, reflection matrix, should support the satellite communication paradigm.

Validity of the findings

In this paper, the authors have highlighted the potential of IRS in optimising satellite connectivity for IoT applications. The intelligent reflecting surfaces are metasurfaces that enable smart radio by introducing controlled reflection. The IRS offers improved communication by optimising the signal reception and transmission. However, the use of IRS for optimising the satellite connectivity has several challenges which are not taken into account in this paper. However, the following concerns need to be addressed before final publication.
1. The writing of the abstract needs to focus on: purpose, methodology, findings, practical implications. It is recommended to rewrite the abstract for better understanding to the readers.
2. In the IoT-satellite communication landscape, how the IRS integration is taking place? Which IRS placement strategy is adopted. Practical things like height of IRS, its placement, orientation, reflection matrix, should support the satellite communication paradigm.
3. In the literature part, authors are recommended to cite the latest references that focus on the integration of IRS for IoT satellite communication regime. Moreover, the utilisation of IRS for various other applications need to be highlighted.

G. Li, M. Zeng, D. Mishra, L. Hao, Z. Ma and O. A. Dobre, "Energy-Efficient Design for IRS-Empowered Uplink MIMO-NOMA Systems," in IEEE Transactions on Vehicular Technology, vol. 71, no. 9, pp. 9490-9500, Sept. 2022, doi: 10.1109/TVT.2022.3177132.
Taneja, A., Rani, S., & Herencsar, N. (2023). Energy aware solution for IRS-aided UAV communication in 6G wireless networks. Sustainable Energy Technologies and Assessments, 58, 103318.

4. It is not clear whether the IRS would be mounted on the satellite (SAT-side) or near the ground station (GN-side) to aide the satellite communication. Which IRS placement approach the authors have used?

5. In results section, fig. 22 “Signal strength Comparative scatter plot” and fig. 24 “Spectral Efficiency Comparative scatter plot”, what is the inference drawn from these scatter plots?

6. To support the IRS aided satellite communication for IoT applications, which channel model is used? And what about IRS phase shift optimization and beamforming? The authors need to elaborate these points in this paper.
7. There are many grammatical errors and semantic ambiguities in this manuscript. The authors are advised to proofread the manuscript very carefully.

Reviewer 3 ·

Basic reporting

In general, the paper is well written. The references cited in the text are relevant for the topic being dicussed. However, there are several apects that should be improved:

1) A proper background on the different types of IRS should be provided in Section 2. Normally, IRS can be classified as: (i) passive RIS with no reconfigurability (their elements present always the same phase shifts, so the IRS can't be reconfigured or programmed dynamically), (ii) nearly-passive IRS where the elements can be reconfigured to adjust their phase shifts dynamically according to the channel response or users' needs (this type of RIS is not passive, as it requires some energy supply to reconfigure the elements), and (iii) active RIS that can amplify the incident signal and are also reconfigurable (this type of RIS requires much more energy to amplify the signals).

2) Figure quality should be improved. Figures from 8 to 31 (showning the results) are screenshots whit low image quality. Instead of using screenshots, please edit properly the plots.

3) In Figures 18, 20, and 22, the same range of values should be used for the colour scale in both right and left plots. This facilitates the comparison of the results for the cases with RIS and without RIS.

4) Some acronyms are defined several times: IRS (lines 77, 132, 145, etc), RIS (lines 80, 145, etc), IoT (lines 35, 62, 101, etc)... Acronyms should be defined only one time, and then, use the acronym as it is.

Experimental design

The weakest part of the paper is the experimental design and research methodology. There are many key aspects that should be clarified:

1) The authors should properly explain what the contributions of their work are. In Section 2, several papers about IRS-assisted uplink for satellite communications are cited. However, there is no information on the contributions of this work. Is there any novel approach? Any unexplored aspect of IRS-assisted communications? Any contribution or complementary study that reinforces the previously reported works in the literature?

2) In general, there are very few details about the IRS definition and operation, which is the key part of the work. For instance, related to comment #1 of the Basic Reporting section: which type of RIS have the authors considered here? It is a nearly-passive IRS that can be reconfigured dynamically?

3) Related to the previous comment: why is N=8 the number of elements of the IRS? Is there any reason for that? In my opinion, N=8 is a very small IRS. Note that the element size is typically around (0.5*wavelength), so an IRS panel typically has several tens or hundreds of elements. Why not using a 30x30-element RIS (N=900)? What criteria has been followed to set N?

4) How are computed the phase shifts (tetha) introduced by the RIS elements? Is there any equation or analytic procedure? This part is of key importance, since the phase shifts of the IRS elements determine the behaviour of the IRS. Have you considered the location of the IoT devices or the satellite to obtain the phase shifts? Have you performed any optimization to obtain the phase shifts? Are the phase shifts dynamically adjusted during the simulation or are they fixed? Different devices located at different positions may require different phase shifts. All these issues should be clearly explained in the paper.

5) Typically, IRS are implemented using flat panels, which are attached to a building wall or other supporting structures. Therefore, the wireless devices that transmit the signals that will be reflected by the IRS cannot be localed at any place. These devices must be in the field of view of the IRS (e.g., in front of the IRS, with a maximum angle of +-90°, but not behind the IRS). Note that IRS are not omnidirectional devices. Have the authors considered the IRS panel orientation and field of view when simulating with the IoT devices? It seems that the simulations are not very realistic, as they consider that the IRS can reflect all the signals, no matter where the device is located.

6) Figure 7 has several differences with respect to its description in section 4.2.1 (lines 425-447): why the link without RIS has no AWGN channel? Where is the Sum Block that adds the x and y signals? The signal reflected by the IRS (y) goes directly to the Gain Block without any sum.

7) Why has the Gain Block (path loss) a value of 1? Why are the Tx and Rx gains equal to 1? There are a lot of parameters and the authors do not explain what criteria has been followed to set the parameter values.

8) What is the frequency of the communication between the IRS and the satellite? Is it 2.4 GHz or 14 GHz? Different frequency parameters are set for each device, and there is no explanation about what is the actual frequency of the uplink that is being simulated.

9) IRS optimization codes should be described in detail in section 4.2.2, since this is the core of the work. The authors describe the benefits of using the IRS, but not how to design or program the IRS to achieve them. If the topic of the work is IRS-assisted communications, the way in which the IRS has been optimized or programmed has to be explained in detail.

10) At the end of section 4.2.2 (Performance metrics), the analytic formulas that are used to compute each metric should be given in the text.

Validity of the findings

The results section requires a deeper analysis of the results achieved:

1) In Fig. 10, some IoT devices (in general, near the edges of the plot) show a higher SNR value without IRS than with IRS. How is this possible?

2) How many IRS are you considering in the simulations ? According to Figs. 10, 12, 14, and so on, it seems that there are 3 IRS instead of 1. Why? This should be clearly explained in the paper (number of IRS and location).

3) In Figs. 12, 14, 24, 26, 28 and 30, again, some devices present a better performance without IRS than with IRS. The reasons should be explained.

Reviewer 4 ·

Basic reporting

(1) Enhance the introduction by clearly stating the research gap this study aims to fill. This will help readers understand the significance of the research.
(2) Revise Figure 7 to ensure it is labeled and described comprehensively. Labels for "With and Without IRS" and "Satellite" should accurately represent the physical model implementation. The system should also include channel models other than AWGN. This includes adding captions that explain the relevance of each figure to the study. I strongly suggest that the system model be revised.
(3) Consider adding a discussion that contrasts the findings of this study with those from other recent works, highlighting what is novel or different about your approach.
(4) The system architecture as well as the system model should be revised. For example, while the authors mention encoding methods like ASK, PSK, PCM, FSK, and QAM, they do not specify which channel coding methods are considered, nor do they mention specific types of modulation and the reasons for their selection.

Experimental design

(1) Enhance the introduction by clearly stating the research gap this study aims to fill. This will help readers understand the significance of the research.
(2) Revise Figure 7 to ensure it is labeled and described comprehensively. Labels for "With and Without IRS" and "Satellite" should accurately represent the physical model implementation. The system should also include channel models other than AWGN. This includes adding captions that explain the relevance of each figure to the study. I strongly suggest that the system model be revised.
(3) Consider adding a discussion that contrasts the findings of this study with those from other recent works, highlighting what is novel or different about your approach.
(4) The system architecture as well as the system model should be revised. For example, while the authors mention encoding methods like ASK, PSK, PCM, FSK, and QAM, they do not specify which channel coding methods are considered, nor do they mention specific types of modulation and the reasons for their selection.

Validity of the findings

(1) This paper has a high similarity index of 83%, with similar work found in the Nottingham Trent University database system. It would be better for the author to cite related references and paraphrase and write in their own words.
(2) The findings, while promising, are based entirely on simulations. This raises questions about how well these results will translate to real-world scenarios, particularly in environments with unpredictable interference or physical obstacles.
(3) The discussion of the results could be more critical, examining cases where the proposed system might not perform as well, such as in highly dynamic environments.
(4) The study lacks a detailed sensitivity analysis, which would help understand how robust the findings are to changes in key parameters like IRS element configuration or IoT device density.

Additional comments

(1) Avoid repeatedly defining abbreviations without using them later. Follow a consistent format, either capital or small letters, like “Channel state information (CSI)”, “reconfigurable intelligent surfaces (RIS)”, and “Intelligent Reflecting Surface (IRS)”.
(2) Many of the same paragraphs are used repeatedly; I suggest using them in a paraphrased manner.
(3) There are some typographical errors.

---

## Round 0.2 · Minor Revisions

The reviewer has suggested some important minor changes that should be made.

Reviewer 3 ·

Basic reporting

The authors have answered some of my comments. However, there are questions of my first review that have not been addressed:

1) In Figures with two plots (e.g., 16, 18, 20, etc), the same range of values should be used for the vertical colour scale in both right and left plots. This facilitates the comparison of the results for the cases with RIS and without RIS.

2) Figure quality should be improved. Most figures are screenshots whit low image quality. Instead of using screenshots, please edit properly the plots in Matlab and then save them, e.g., as JPG (in the figure window in Matlab -> File -> Save as).

Experimental design

Again, the authors have made an effort to answer several of my comments. However, there are important questions of my first review that have not been addressed (the details about the IRS phase-shifts and the optimization algorithm in section 4.2.2 are still very vague). Please, provide a clear explanation for readers about the following points:

1) How are computed the phase shifts (or the reflection coefficients) of the RIS elements? Is there any equation or analytic procedure?

2) Have you considered the location of the IoT devices or the satellite to obtain the phase shifts? Or you just try randomly different phase-shifts to find those that most improve the system performance?

3) The sentences included in Section 4.2.2 about the IRS optimization are also very general. How does the reinforcement learning algorithm works? Which kind of optimization technique is used?

Validity of the findings

Most of my comments have been answered by the authors, but there is still a question unsolved from my first review:

1) How many IRS are you considering in the simulations of section 5? It seems that there are 3 IRS instead of 1 IRS. Why this choice? This should be clearly explained in the paper (number of IRS and location).

Additional comments

In this revised version, the authors have made a significant effort to improve the paper and address several of my comments. However, there are some important questions of my first review that have not been answered (see my comments in the previous sections). I think that these questions should be answered to reach a final version of the paper that can be accepted for publication.

---

## Round 0.3 · accepted · Accept

The authors have made all the requested changes

Reviewer 3 ·

Basic reporting

In this revised version, the authors have addressed all my previous comments. I do not have any additional recommendations.

Experimental design

The authors have addressed my questions and relevant information about the computation of the IRS phase-shifts and IRS optimization has been included in the revised manuscript. The new content increases the quality of the paper.

Validity of the findings

The authors have provided additional information about the simulation parameters that was missing in the original manuscript. I do not have any further comments.

Additional comments

I think that the revised paper can be accepted for publication as it is.